# Recent Advances in the Clinical Targeting of Hedgehog/GLI Signaling in Cancer

**DOI:** 10.3390/cells8050394

**Published:** 2019-04-29

**Authors:** Hao Xie, Brooke D. Paradise, Wen Wee Ma, Martin E. Fernandez-Zapico

**Affiliations:** 1Division of Medical Oncology, Mayo Clinic, Rochester, MN 55905, USA; xie.hao@mayo.edu (H.X.); ma.wen@mayo.edu (W.W.M.); 2Schulze Center for Novel Therapeutics, Division of Oncology Research, Mayo Clinic, Rochester, MN 55905, USA.; paradise.brooke@mayo.edu

**Keywords:** Hedgehog, Smoothened, GLI factors, clinical trials, vismodegib, sonidegib

## Abstract

The Hedgehog/GLI signaling pathway plays an important role in normal embryonic tissue development and has been implicated in the pathogenesis of various human cancers. In this review article, we summarize pre-clinical evidence supporting the suitability of targeting this signaling pathway in cancers. We review agents blocking both the ligand-dependent and ligand-independent cascades, and discuss the clinical evidence, which has led to the FDA approval of Hedgehog receptor Smoothened inhibitors, vismodegib, and sonidegib, in different malignancies. Finally, we provide an overview of published and ongoing clinical trial data on single agent or combination therapeutic strategies, targeting Hedgehog/GLI signaling pathway, in both advanced solid tumors and hematologic malignancies.

## 1. Overview of Hedgehog Signaling Pathway

The Hedgehog (Hh) signaling pathway plays an important role during normal embryonic tissue development. It precisely controls epithelial and mesenchymal cell interactions regulating cell proliferation and differentiation [1,2]. The Hh pathway is comprised of four main components: a 12-transmembrane protein, patched-1 (PTCH1), a 7-transmembrane protein, smoothened (SMO), cytoplasmic signaling effectors (i.e., Suppressor of Fused (SUFU)), and a family of downstream transcription factors, glioma-associated oncogene homolog, (GLI1-3) [1]. As shown in Figure 1, in the absence of extracellular ligand (Indian Hh, Desert Hh, and Sonic Hh) binding, PTCH1 constitutively suppresses Hh pathway activation by inhibiting SMO [3]. When extracellular ligand binds PTCH1, the receptor is internalized from cell surface and degraded, releasing its suppression of SMO. Active SMO then promotes the translocation of activated GLI proteins into the nucleus and induces expression of target genes including cyclin-dependent kinases and growth factors, ultimately promoting cell growth, survival and differentiation [4].

The Hh pathway can be aberrantly activated through various mechanisms, which can lead to the pathogenesis of numerous human cancers. One of the most common ones is the Hh ligand upregulation in autocrine or paracrine fashion; this was reported in pancreatic, gastrointestinal, breast, lung, medulloblastoma, and prostate cancers [4,5]. Other mechanisms include, the loss-of-function mutations in the signaling repressors like PTCH1; loss-of-function mutations in the GLI1 inhibitory protein, SUFU; and, less commonly, gain-of-function mutations of SMO [1]. Some of these genetic aberrations are the drivers of the heritable basal-cell nevus syndrome (Gorlin syndrome), in which patients inherit only one functional copy of genes encoding PTCH1. This leads to the development of numerous skin basal cell carcinomas (BCC) and an increased risk of rhabdomyosarcoma and medulloblastoma [5]. Not only can the activating mechanisms contribute to the development of tumors, but they have also been implicated in the emergence of resistance (i.e., lymphoma, multiple myeloma, and chronic myeloid leukemia (CML)) [6].

## 2. Pre-Clinical Evidence on Targeting Hedgehog Signaling Pathways

Although reinforcing endogenous negative regulation of Hh signaling, via PTCH1 and SUFU could be effective, current efforts center on developing new and refining existing inhibitors against SMO. In addition, efforts have been made to inhibit GLI transcription factors to tackle, for example, acquired resistance SMO to small molecule inhibitors via activating mutations or noncanonical Hh pathway activation. These efforts have culminated two FDA approved Hh pathway inhibitors in the clinic. Examples of pre-clinical evidence on targeting Hh signaling pathways are discussed below.

### 2.1. Ligand-Dependent Hh Signaling Inhibition

Canonical activation of the Hh signaling pathway is most often inhibited by targeting SMO. Hh pathway inhibition, with small molecules, has been shown to decrease tumor growth in different models of various cancer types. For example, SMO inhibitor, saridegib, has been successfully used in combination with chemotherapy followed by maintenance therapy to suppress serous ovarian cancer xenograft growth [7]. Vismodegib, another SMO inhibitor, was effective both, as a single agent and in combination with standard chemotherapy agents, in primary colorectal cancer xenografts [8]. In chondrosarcoma, pre-clinical findings demonstrated that SMO inhibitors significantly decreased tumor size and cellularity in human chondrosarcoma xenograft [9]. Hh signaling pathway is dormant in adult pancreas but activated in approximately 70% of pancreatic adenocarcinomas [10,11]. Pre-clinical models have demonstrated that aberrant Hh pathway activation, through either, ligand-dependent or ligand-independent mechanisms, plays key roles in both the initiation and metastasis of pancreatic adenocarcinomas [10,11,12,13,14,15,16]. Pre-clinical studies in transgenic mouse model showed that SMO inhibitors, such as saridegib were able to reduce the density of desmoplastic stroma in pancreatic cancer tissue and increase gemcitabine delivery to the tumor, due to increased intra-tumoral mean vessel density [13]. As a result, decreased tumor growth and metastases, and increased survival of transgenic mice, were observed [17].

Similar to the FDA approved vismodegib and sonidegib, glasdegib is a potent, selective small molecule inhibitor of SMO [18]. In addition to its significant anti-tumor activity in vivo, glasdegib demonstrated its ability to reduce the number of leukemic stem cells in mouse xenograft [19]. Glasdegib has been extensively evaluated in hematologic malignancies, such as acute myeloid leukemia (AML), myelodysplastic syndrome (MDS), and myelofibrosis, and demonstrated anti-tumor activities in combination with non-intensive chemotherapy regimens [20].

Saridegib, a semi-synthetic derivative of cyclopamine, is structurally distinct, compared to the previously discussed small molecule inhibitors. Saridegib has improved selectivity, affinity, and stability compared to cyclopamine. It is a potent and selective SMO inhibitor with an IC50 of 1.4 nM [21]. In a medulloblastoma allograft model, saridegib demonstrated dose-dependent inhibition of Hh pathway via downregulation of GLI1 mRNA expression, which correlated with prolonged survival [22]. In a transgenic pancreatic cancer model, saridegib disrupted interactions between tumor cells and expanded stroma by inhibiting the paracrine mechanism of Hh pathway activation. As a result, intra-tumoral concentration of gemcitabine significantly increased [13].

TAK-441 and BMS-833923 are other small molecule SMO inhibitors under clinical investigation. In preclinical studies, TAK-441 has a high anti-tumor activity with an IC50 of 4.4 nM in multiple tumor types [23,24]. TAK-441 has activity against, not only Hh ligand overexpression via autocrine and paracrine mechanisms, but also mutation-driven Hh signaling pathway activation [24].

Itraconazole, an FDA-approved antifungal drug, was discovered from screening a library of existing drugs for its ability to inhibit angiogenesis and Hh signaling pathway by antagonizing SMO [25]. Recent studies have demonstrated moderate success in repurposing this anti-fungal as an anti-cancer agent. In a mouse medulloblastoma model, with a constitutive activation of Hh signaling pathway, itraconazole effectively downregulated GLI1 expression [26]. It has been evaluated clinically in prostate cancer and non-small cell lung cancer [25,27]. The clinical benefit of high dose itraconazole in prostate cancer was found to be derived from Hh signaling pathway inhibition instead of antiandrogen effect [25].

### 2.2. Ligand-Independent Hh Signaling Inhibition

The terminal effectors in the Hh signaling pathway, GLI1-3, can be activated through alternate mechanisms, apart from SMO activation. This ligand-independent pathway activation has been difficult to target previously, however, recent advances have been effective in directly inhibiting GLI factors, ultimately dampening the effects of active Hh signaling.

In contrast to anti-SMO agents, GLI inhibitors like arsenic trioxide inhibits the Hh signaling pathway by preventing the accumulation of GLI2 and reducing the transcriptional induction of target genes of GLI2. In a mouse model of medulloblastoma, with Ptch+/− p53−/−, arsenic trioxide inhibited tumor growth at a similar serum level achieved in the treatment of acute promyelocytic leukemia [28]. In a xenograft model of Ewing sarcoma, arsenic trioxide was found to inhibit tumor cell growth by direct GLI1 binding and inhibition of its transcriptional activity [29]. In a mouse model of drug-resistant SMO(D477G) medulloblastoma, arsenic trioxide alone or in combination with itraconazole, was able to effectively inhibit tumor growth in vivo [30]. GANT61 is another selective GLI inhibitor, which significantly downregulates the transcriptional activity of GLI1 and GLI2 by blocking their DNA binding. GANT61 inhibits pancreatic cancer stem cell growth in vitro and in vivo [31], in addition to its antitumor activity alone, or in combination in multiple other tumor types [32,33,34].

## 3. Clinical Trials Using Inhibitors of the Hedgehog Signaling Pathway

Various small molecule SMO inhibitors, such as vismodegib and sonidegib, have been chosen as initial targets for the clinical evaluation of Hh pathway inhibition, due to their improved pharmacological properties from pre-clinical development and predictable Hh signaling pathway activation in BCC. Previous clinical trial results, and ongoing clinical trials evaluating Hh signaling pathway inhibitors, are summarized in Table 1, and Table 2, respectively.

### 3.1. Advanced Solid Tumors

Similar to other targeted therapies, a number of Hh inhibitors have been evaluated initially in advanced solid tumors. In a phase 1, first-in-human trial conducted with patients from western countries, sonidegib at doses from, 100 to 3000 mg daily or 250 to 750 mg twice daily, were administered to 103 patients with advanced solid tumors [35]. A similar phase 1 trial of sonidegib was done in Asian patients [36]. In both trials, grade 3/4 creatine kinase elevation with rhabdomyolysis were identified as dose-limiting toxicity (DLT). The western population of patients had significant grade 3/4 gastrointestinal toxicities patients whereas the Asian population had grade 3/4 abnormal liver function tests (LFT). It was noted that East Asian patients have lower tolerability with recommended dose (RD) of 400 mg daily compared to Western counterparts with maximum tolerated dose (MTD) of 800 mg daily and 250 mg twice daily. Clinical responses varied from no response to complete response, however in this proof-of-concept study, biomarker correlation studies were prioritized, and showed a significant association between Hh signaling pathway activation and clinical response. Patients with more sonidegib exposure had more reduction in GLI1 mRNA expression in the tumor compared with matching normal skin samples [35]. In a phase 1 trial of patients with advanced solid tumors, Saridegib demonstrated higher liver toxicities when compared to sonidegib with up to 66% grade 3/4 LFT elevations [37]. Vismodegib, glasdegib, and TAK-441, in phase 1 trials of patients with advanced solid tumors [38,39,40], had not only significantly fewer grade 3/4 toxicities but also different types of toxicities compared to sonidegib [36] and saridegib [37]. Both vismodegib and TAK-441 had 12% grade 3/4 hyponatremia and less than 10% of other grade 3/4 adverse events [38,40]. Phase 1 trial of vismodegib did not observe any DLT with MTD not reached [38]. Similarly, grade 2/3 DLTs from glasdegib were only observed at the highest dose level of 640 mg daily [39]. Regarding preliminary activities of these agents, approximately 30% of patients achieved partial response (PR) or stable disease (SD). The majority were patients with BCC or medulloblastoma, consistent with previous findings [35].

### 3.2. Small Cell Lung Cancer

Sonidegib was evaluated in a phase 1 trial in 15 patients with extensive stage small cell lung cancer (SCLC) [41]. Sonidegib at two dose levels (400 mg and 800 mg) was administered in combination with standard dose of cisplatin and etoposide. 200 mg daily dose de-escalation was allowed. MTD of sonidegib was 800 mg daily. Grade 3/4 nausea and febrile neutropenia were considered DLT. 79% of patients had PR from this regimen. 

Vismodegib in combination with cisplatin and etoposide was compared to standard cisplatin and etoposide for patients with extensive stage SCLC in a randomized phase 2 trial [42]. Fifty-two patients were randomized to receive vismodegib 150 mg daily with chemotherapy. Grade 3-5 adverse events, mainly hematologic toxicities, progression-free survival (PFS), as the primary end point, were similar across all arms.

### 3.3. Basal Cell Carcinoma

Vismodegib and sonidegib were FDA-approved for patients with locally advanced and metastatic BCC in 2012, and 2015, respectively. A phase 1 study on vismodegib, in patients with locally advanced and metastatic BCC, determined the recommended dose for phase 2 was 150 mg daily. Vismodegib was very well-tolerated with only grade 3/4 fatigue and hyponatremia. No DLT was observed. Fifty four percent of patients had disease response and another 33% had stable disease. Tissue correlation studies demonstrated upregulation of Hh signaling pathway in 96% of patients as determined by elevated GLI1 mRNA expression. Vismodegib exposure led to a significant decrease in GLI1 expression [43]. The phase 2 registration trial of vismodegib for patients, with advanced and metastatic BCC, not only confirmed the favorable safety profile, but also showed 30% overall response rate (ORR) in metastatic BCC and 43% ORR, including 21% complete response (CR) in locally advanced BCC [44]. Long-term follow-up reported 48.5% ORR in metastatic BCC and 60.3% ORR in locally advanced BCC, with no additional long-term or delayed toxicities. Median overall survival (OS) was 33.4 months in metastatic BCC and not reached in locally advanced BCC [45]. The aforementioned findings on vismodegib in BCC were confirmed in a large open-label safety trial, with 1215 patients, which is representative of real clinical practice setting [46,47]. Vismodegib was also evaluated in two small phase 2 trials for resectable BCC in the neo-adjuvant setting. Continuous neoadjuvant vismodegib for at least 3 months, followed by surgery was able to significantly reduce surgical defect area [48] and achieve adequate complete histologic clearance rate [49]. In a phase II trial, patients with basal-cell nevus syndrome were randomized to vismodegib 150 mg daily versus placebo. Vismodegib arm had significantly decreased number of new BCC cases per year (2 versus 29) and decreased size of existing multiple BCC lesions, compared to placebo arm (−65% versus −11%). However, vismodegib discontinuation rate was 54%, due to adverse events [50,51]. For this reason, and the need for long-term vismodegib in patients with multiple BCCs, two intermittent dosing strategies were evaluated in 229 patients and was found to have similar activity and good tolerability, while on the drug compared to historical continuous dosing [52].

As for other agents in the phase 2 registration trial, patients with advanced or metastatic BCC were randomized into sonidegib at two dose levels: 200 mg daily or 800 mg daily. Sonidegib at 200 mg daily had lower grade 3/4 toxicities and lower dose interruption, reduction or discontinuation. Both dose levels provided equivalent ORR, comparable to that of vismodegib from previous reports [53]. A phase II open-label trial evaluated the activity of itraconazole in patients with BCC. Significant reductions of cell proliferation (Ki67), Hh signaling pathway activity (GLI mRNA), and tumor size were observed [54].

### 3.4. Medulloblastoma

The activity of vismodegib, in refractory metastatic medulloblastoma, was first reported in a patient whose tumor harbored a somatic mutation of PTCH1, thus activating the Hh signaling pathway [3]. Phase 1 trial of vismodegib, in pediatric patients with recurrent or refractory medulloblastoma, established RD of 150 to 300 mg daily, based on body surface area. DLT included grade 3 LFT elevation, thrombocytopenia, and grade 4 hypokalemia [55]. Subsequent phase 2 trial included both adult and pediatric patients with vismodegib 150 mg daily. Significant grade 3/4 adverse events (AEs) were lymphopenia, seizure, hypokalemia, and headache. An objective response was only observed in Hh-subgroup with ORR 15%. Among them, loss of heterozygosity of PTCH1 predicted longer PFS [56]. Sonidegib was also evaluated in relapsed medulloblastoma in a phase 1/2 trial, with both adult and pediatric patients. Adult patients received sonidegib 800 mg daily and were observed to have grade 3/4 creatine kinase and LFT elevations. Fifty percent of the patients with activated Hh pathway in their tumor had disease response to sonidegib, which was translated to longer disease-free survival [57]. 

### 3.5. Pancreatic Adenocarcinoma

Hh pathway inhibitors, including vismodegib and saridegib, have not demonstrated clinical benefit in addition to standard chemotherapy for metastatic pancreatic adenocarcinoma. Vismodegib was initially assessed for its effect on Hh pathway inhibition, cancer stem cells, cell proliferation, fibrosis, and clinical benefit. Although vismodegib effectively inhibited Hh signaling pathway activity, its addition to gemcitabine did not show additional clinical benefit [58]. This was further confirmed in a larger randomized phase 2 trial to compare vismodegib, plus gemcitabine, with gemcitabine alone. There was no statistically significant difference in terms of ORR, PFS, and OS between these two arms [59]. Saridegib in combination with 5-FU, leucovorin, irinotecan, oxaliplatin (FOLFIRINOX), in a phase 1 trial determined the MTD at 130 mg daily. This demonstrated similar toxicity profile and ORR, but was halted early, due to a similar unpublished phase 2 trial demonstrating the detrimental effect of this combination regimen [60].

### 3.6. Other Solid Tumors

Vismodegib was evaluated in clinical trials for a few other types of solid tumors, but it has not demonstrated additional clinical benefit. For example, vismodegib, in a phase I trial for patients with metastatic castration-resistant prostate cancer, showed effective Hh pathway inhibition, but no clinical response [61]. Vismodegib was also used as a maintenance therapy for patients whose ovarian cancer achieved second or third complete response after initial chemotherapy. It showed similar median PFS as compared with that from placebo arm [62]. In a phase 2 trial, patients with metastatic colorectal cancer were randomized to vismodegib versus placebo, with a standard chemotherapy regimen. Vismodegib arm had an equivalent ORR and median PFS, compared to the placebo arm [63]. Vismodegib was also combined with 5-FU, leucovorin, oxaliplatin (FOLFOX) in a phase 2 randomized trial of patients, with locally advanced or metastatic gastric or gastroesophageal junction adenocarcinoma. Adding vismodegib to FOLFOX did not increase grade 3-5 toxicities, nor did it improve the response rate or survival either, when compared to FOLFOX alone [64]. In a phase 2 trial of patients with advanced chondrosarcoma, vismodegib 150 mg daily did not meet predefined 6-month clinical benefit rate of 40% [65]. Vismodegib in combination with a γ-secretase/notch inhibitor RO4929097 was evaluated in advanced sarcoma in a phase 1b trial. The combination was well-tolerated, but only stable disease was observed [66]. Patients with recurrent glioblastoma multiforme, in a phase 2 trial, were randomized to pre-operative and post-operative vismodegib arm versus post-operative vismodegib arm, with the idea that Hh pathway inhibitors selectively target glioma cancer stem cells. Although a significant decrease in the number of CD133+ neurospheres was observed in pre-operative and post-operative vismodegib arm, vismodegib did not appear to be clinically active in recurrent glioblastoma multiforme [67].

### 3.7. Hematologic Malignancies

Glasdegib was evaluated in phase 1 trials conducted in both Japanese [68] and western patients [69], with hematologic malignancies including, AML, MDS, CML, chronic myelomonocytic leukemia (CMML), and myelofibrosis. The recommended dose for phase 2 was 100 mg daily. DLTs were observed only at high doses. Common grade 3/4 AEs included thrombocytopenia, hypokalemia, pyrexia, and anorexia. Moderate response rates were observed in various hematologic malignancies [68,69]. Single agent glasdegib was subsequently evaluated in a small phase 2 trial for patients with refractory MDS and CMML. Glasdegib was well tolerated but had limited single agent activity [70]. As a result, glasdegib was subsequently combined with chemotherapy as a frontline therapy for patients with high risk MDS and AML. In a phase 1b trial, glasdegib was combined with low-dose cytarabine, decitabine, or cytarabine/daunorubicin, based on patient eligibility for intensive chemotherapy. No DLT was observed in non-intensive chemotherapy arm. Grade 4 neuropathy was DLT in intensive chemotherapy arm. In a subsequent randomized phase 2 trial, glasdegib 100 mg plus low-dose cytarabine was compared with low-dose cytarabine alone, in patients with high risk MDS and AML who were not eligible for intensive chemotherapy [71]. The glasdegib arm had a significantly higher complete response rate (15% versus 2%) and longer median OS (8.3 versus 4.9 months). Based on this phase II finding, FDA granted priority review on glasdegib in patients with previously untreated AML [71]. Other Hh pathway inhibitors, such as saridegib and vismodegib were evaluated in phase 2 trials for patients with myelofibrosis and non-Hodgkin lymphoma (NHL), or chronic lymphocytic leukemia (CLL), respectively. However, no significant clinical activity was observed [72,73].

## 4. Conclusions and Future Directions

Currently, there are only two FDA-approved Hh pathway inhibitors, Vismodegib and Sonidegib, which are limited to locally advanced and metastatic BCC. Recently, Glasdegib was granted priority review by the FDA in patients with previously untreated AML, in order to increase the number of inhibitors available to patients to three [71]. Hh pathway inhibitors, in general, have limited single agent activity in unselected patients, with multiple types of cancers in early phase clinical trials. In addition, acquired resistance to SMO inhibitors [3,75] and cross-resistance to different types of SMO inhibitors [76,77] have been observed in patients, whose disease had initial response to therapy and then experienced disease relapse. Genomic studies in mouse models and human tumor biopsy specimens revealed five major mechanisms of resistance: 1) SMO mutations that directly impair drug binding to SMO binding pocket, for example, a de novo SMO D473H missense mutation in medulloblastoma [78,79]; 2) SMO mutations that constitutively activate Hh signaling pathway independent to SMO inhibitor binding [80]; 3) copy number variations or mutations of SUFU or GLI2 [76]; 4) intra-tumoral and inter-tumoral heterogeneity of Hh signaling pathway activity [80]; 5) ligand-depend cancer and stroma interactions, such as in pancreatic adenocarcinoma [60]. Other resistance mechanisms of SMO inhibition involve activation or upregulation of other signaling pathways, that directly affect GLI activities, such as PI3K-mTOR, aPKC-ɩ/λ, BRD4, and PDE4 signaling [78,81,82,83,84]. These at least partially explain why some Hh inhibitors showed significant clinical activity, while others did not.

Current and future research efforts in targeting Hh signaling in cancer should focus on strategies addressing and overcoming these resistance mechanisms. The design and development of next generation therapeutics, targeting Hh signaling, should take into consideration both, acquired SMO mutations and downstream genetic variants, such as SUFU loss-of-function mutations and GLI gain-of-function mutations. To expand the scope of Hh signaling pathway regulation, GLI inhibitors such as arsenic trioxide and GANT61 have been evaluated in both pre-clinical and clinical settings [28,31]. To address the limited single-agent activity of Hh pathway inhibitors, combination therapy, concurrently targeting other upregulated signaling pathways, may be promising. For example, a combination of sonidegib and PI3K inhibitor buparlisib is currently under evaluation for patients with advanced solid tumors [85]. Along the same line, the strategy of using a single gene expression panel for patient selection, as used in previous medulloblastoma studies, is no longer valid in the setting of complex signaling responses, and a well-recognized presence of tumor heterogeneity. Future efforts should focus on developing more comprehensive tools to evaluate complex signaling processes. Such tools will significantly improve precise stratification of patients with tumor subtypes likely respond to therapy, as a result, maximize patient outcomes [81].

## Figures and Tables

**Figure 1 cells-08-00394-f001:**
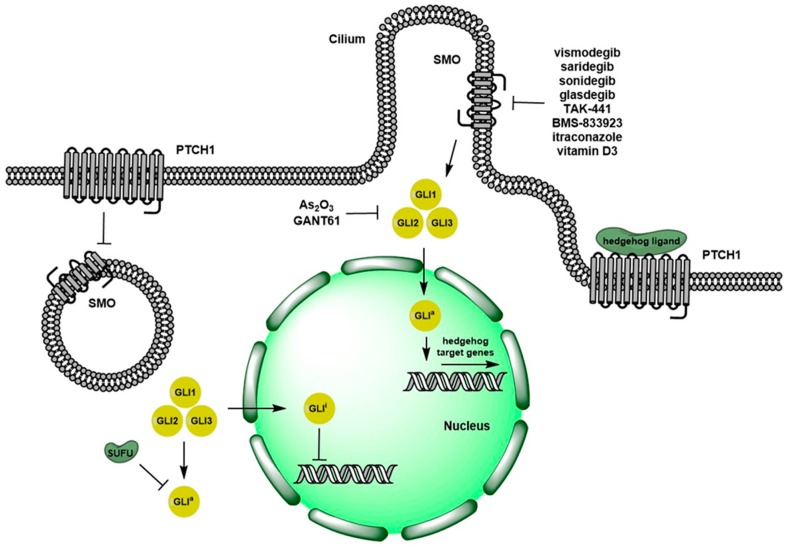
Schematic of Hedgehog signaling pathway. In the absence of Hh ligand binding, the receptor PTCH1 acts as a negative regulator of Hh pathway by inhibiting SMO. When Hh ligand binds to PTCH1, it releases the inhibition of SMO. Then, SMO initiates a downstream signaling cascade leading to the activation of GLI transcription factors, which translocate to the nucleus and induce Hh pathway target gene expression. GLIa: activated GLI. GLIi: inactivated GLI.

**Table 1 cells-08-00394-t001:** Published clinical data with hedgehog signaling pathway inhibitors in the past five years.

Disease Type	Clinical Trial Phase (# Patients)	Dosing and Schedule	Adverse Events (G3-5 ≥ 10%)	Activities
Extensive stage SCLC	II (152) [42]	Cisplatin/etoposide q3w, with or without vismodegib 150 mg daily x 4 cycles	G3-5 neutropenia (53%), febrile neutropenia (12%)	ORR 56%, PFS 4.4 months, OS 9.8 months, similar to cisplatin, etoposide arm
	I (15) [41]	Cisplatin/etoposide q3w, sonidegib 400 mg and 800 mg daily (MTD: 800 mg)	G3/4 anemia (33%), neutropenia (53%), CK elevation (13%), fatigue (13%), nausea (13%). DLT: nausea, febrile neutropenia	PR: 79%
Advanced solid tumors	I (103) [35]	Sonidegib 100 to 3000 mg daily and 250 to 750 mg twice daily (MTD: 800 mg daily and 250 mg twice daily)	G3/4 nausea (25%), dysgeusia (29%), anorexia (29%), muscle spasms (32%), fatigue or asthenia (27%). DLT: G3/4 CK elevation (18%)	CR/PR: 37% for BCC, 33% for medulloblastoma; SD: 23%
	I (45) [36]	Sonidegib 400 to 800 mg daily (RD: 400 mg daily)	G3/4 elevated LFT (15%). DLT: G3/4 CK elevation (24%), rhabdomyolysis (10%)	SD: 33%
	I (94) [37]	IPI-926 20 to 210 mg daily (RD: 160 mg daily)	G3/4 anemia (18%), elevated LFT (66%), fatigue (37%). DLT: G3 LFT elevation, fatigue, anorexia	ORR 29% in BCC cohort
	I (23) [39]	Glasdegib 80 to 640 mg daily (MTD: 320 mg daily)	DLT: G2 fatigue, hypotension and G3 nausea, vomiting, dehydration at 640 mg daily	SD: 35%
	I (34) [40]	TAK-441 50 to 1600 mg daily (MTD: 1600 mg daily)	G3/4: hyponatremia (12%), DLT: muscle spasms and fatigue	PR: 3%, SD: 21%
Advanced or metastatic BCC or Basal-cell nevus syndrome or Resectable BCC	II (230) [53]	Sonidegib 200 mg vs. 800 mg daily	G3/4 elevated CK (13% in 800 mg arm)	ORR: 36% in 200 mg arm, 34% in 800 mg arm
	II (41) [50,51]	Vismodegib 150 mg daily vs. placebo for 18 months	G3/4 weight loss (15%)	New surgically eligible BCC: 2 (vismodegib) vs. 29 (placebo) cases per year
	II (229) [52]	vismodegib 150 mg daily x 12 wks, then placebo x 24 wks, then 150 mg daily x 12 wks (arm A) vs. vismodegib 150 mg daily x 24 wks, then placebo x 24 wks, then 150 mg daily x 8 wks (arm B)	G3/4 muscle spasm (4% in arm A,11% in arm B)	Number of lesion reduction: 63% in arm A, 54% in arm B.
	II (1215) open-label safety trial [46,47]	Vismodegib 150 mg daily		ORR: 68% in locally advanced BCC, 37% in metastatic BCC
	II (15), neoadjuvant [48]	Vismodegib 150 mg daily x 3-6 months before surgery		Surgical defect area reduction: 27%
	II (24) before surgery [49]	Vismodegib 150 mg daily before surgery for cohort 1: 12 wks, cohort 2: 12 wks, then 24 wks observation, cohort 3: 8 wks on, 4 wks off, 8 wks on	Most frequent adverse events: muscle spasms (76%), alopecia (58%), and dysgeusia (50%).	Complete histologic clearance: 42% for cohort 1, 16% for cohort 2, 44% for cohort 3.
	II (29) open-label [54]	Itraconazole oral 200 mg twice daily x 1-month vs. 100 mg twice daily x 2.3 months	G4 congestive heart failure	cell proliferation reduction: 45%, Hh activity reduction: 65%, tumor size reduction: 24%
Recurrent or refractory medulloblastoma	I/II (55) [57]	Adult: Sonidegib 800 mg daily, pediatric: 680 mg/m^2^ (RD)	In adult: G3/4 elevated CK (31%), elevated LFT (12%)	ORR: 50% in patients with activated Hh pathway
	I (33) [55]	Vismodegib 85 to 170 mg/m2, revised to 150 and 300 mg daily (RD)	DLT: G3 γ-glutamyl transferase elevation, thrombocytopenia, G4 hypokalemia	One patient with SHH- subgroup had response
	II (43) [56]	Vismodegib 150 mg daily	G3/4 lymphopenia (30%), seizure (12%)	No response in non-SHH-subgroup. 15% in adult patients with SHH-subgroup, 41% with prolonged disease stabilization
Advanced or metastatic pancreatic adenocarcinoma	I (15) [60]	FOLFIRINOX and IPI-926 130 to 160 mg daily (MTD: 130 mg daily)	G3/4 infection (13%), thrombocytopenia (13%), DLT: G3 elevated LFTs (20%)	ORR: 67%
	Ib/II (113) [59]	Gemcitabine with or without vismodegib 150 mg daily	G3-5 neutropenia (28%), fatigue (13%), thrombocytopenia (11%)	Similar ORR, PFS, and OS
	I (25) [58]	Vismodegib 150 mg daily x 3 wks, then vismodegib + gemcitabine	G3 anemia (12%), LFT elevation (12%)	GLI1 inhibition: 96%, PTCH1 inhibition: 83%, ORR 22%, disease control rate: 65%.
Metastatic castration-resistant prostate cancer	I (9) [61]	Vismodegib 150 mg daily x 4 wks	G3/4 anemia (11%), dehydration (11%), dyspnea (11%), pain (22%), pneumonia (11%), vomiting (11%)	GLI1 inhibition: 57% in tumor, 75% in normal skin. No response. Median PFS 1.9 months, OS 7.0 months
	II (46) [25]	Itraconazole 200 mg vs. 600 mg daily	G3 (600 mg arm) hypokalemia (10%)	PSA PFS at 24 weeks (200 vs. 600 mg): 12% vs. 48%.
Metastatic colorectal cancer	II (199) [63]	Vismodegib 150 mg daily or placebo with FOLFOX or FOLFIRI and bevacizumab	G3-5 neutropenia (22%), diarrhea (12%), nausea (10%), fatigue (18%), weight loss (10%), dehydration (12%)	median PFS HR 1.25 (*p* = 0.3), ORR 46% vs. 51% for vismodegib vs. placebo
Advanced chondrosarcoma	II (45) [65]	Vismodegib 150 mg daily		6-month clinical benefit rate 25.6%. median PFS 3.5 months
Advanced gastric or GEJ adenocarcinoma	II (124), [64]	FOLFOX with or without vismodegib 150 mg daily	G3-5 neutropenia (83%), neuropathy (32%), fatigue (25%), thrombosis (23%), anemia (17%), GI bleeding (13%), hypokalemia (17%), nausea (13%).	ORR 58%, median PFS 7.3 months, OS 11.5 months.
Lung adenocarcinoma	II (23), 2^nd^-line setting [27]	Pemetrexed with or without itraconazole 200 mg daily (stopped early due to 1^st^-line pemetrexed	G3/4 (itraconazole arm) lymphopenia (20%)	PFS at 3 months (itraconazole vs. no itraconazole): 67% vs. 29%
Hematologic malignancies or myelofibrosis	I (13), Japanese patients [68]	Glasdegib 25 to 100 mg daily (RD: 100 mg daily)	G3-4 thrombocytopenia (23%), hypokalemia (15%), DLT: none	AML: CR 8%, SD 31%; MDS: CR 8%, SD 16%.
	I (47) [69]	Glasdegib 5 to 600 mg daily (MTD: 400 mg daily, RD: 200 mg daily or lower)	G3-4 anorexia (11%) DLT: G3 hypoxia, pleural effusion, peripheral edema	CML: partial cytogenetic response 20%; MDS/CMML: SD 57%; myelofibrosis: improvement 29%; AML: ORR 32%, SD 25%
	II (14) [72]	IPI-926 160 mg daily	G3-4 bilirubin elevation (21%)	<50% spleen size reduction: 86%; 64% had no response
NHL and CLL	II (31) [73]	Vismodegib 150 mg daily	G3-5 29%	Indolent lymphoma: (17%)
AML and high risk MDS	II (35) [70]	Glasdegib 100 mg daily x 4 months, 200 mg daily allowed for SD	G3-4 infection (11%)	ORR: 6%; SD: 54%; median OS: 10.2 months
	Ib (52) [74]	Glasdegib 100 or 200 mg daily with low-dose cytarabine (arm A) or decitabine (arm B) or cytarabine/daunorubicin (arm C). RD: 100 mg daily	G3-4 febrile neutropenia (A: 39%, C: 54%), fatigue (A: 22%), neutropenia (A: 22%, B: 57%), anemia (B: 29%), thrombocytopenia (A: 30%, B: 43%), pyrexia (C: 18%). No DLT in arms A, B, grade 4 neuropathy in arm C.	Arm A: CR 8.7% Arm B: CR 29% Arm C: CR 54%
	II (132), ineligible for intensive chemotherapy [71]	Glasdegib 100 mg daily and low-dose cytarabine versus low-dose cytarabine alone	Glasdegib arm: more frequent febrile neutropenia.	Glasdegib + cytarabine vs. cytarabine: CR 15% vs. 2%; median OS: 8.3 vs. 4.9 months

DLTs: dose limiting toxicities; MTD: maximum tolerated dose; RD: recommended dose; ORR: overall response rate; CR: complete response; PR: partial response; SD: stable disease; PFS: progression-free survival; OS: overall survival; G: grade; CK: creatine kinase; LFT: liver function tests; SHH: sonic hedgehog; PSA: prostate-specific antigen; FOLFIRINOX: 5-FU, leucovorin, oxaliplatin, irinotecan; FOLFOX: 5-FU, leucovorin, oxaliplatin; FOLFIRI: 5-FU, leucovorin, irinotecan; SCLC: small cell lung cancer; BCC: basal cell carcinoma; GEJ: gastroesophageal junction; AML: acute myeloid leukemia; MDS: myelodysplastic syndrome; CMML: chronic monocytic leukemia; NHL: non-Hodgkin lymphoma; CLL: chronic lymphocytic leukemia. Wks: weeks.

**Table 2 cells-08-00394-t002:** Ongoing clinical trials evaluating agents targeting hedgehog signaling pathway.

Agent	Tumor Types	Phase of Development	Clinicaltrials.Gov Identifier
BMS-833923	Advanced or metastatic cancer	I	NCT00670189
	Extensive stage small cell lung cancer	I: carboplatin, etoposide and BMS-833923	NCT00927875
	Metastatic gastric, gastroesophageal, or esophageal adenocarcinomas	I: BMS-833923, cisplatin and capecitabine	NCT00909402
Itraconazole	Esophageal cancer	I	NCT02749513
	Prostate cancer	II	NCT01787331
	Skin basal cell carcinoma	I	NCT02735356
	Non-small cell lung cancer	II: itraconazole and chemotherapy	NCT03664115
	Non-small cell lung cancer	I: neoadjuvant setting	NCT02357836
	Basal cell carcinoma	II: SUBA-Itraconazole	NCT02354261
	Various tumors	I: volasertib and itraconazole	NCT01772563
	Glioblastoma	I: itraconazole and temozolomide	NCT02770378
Saridegib	Recurrent head and neck cancer	I: saridegib and cetuximab	NCT01255800
	Metastatic solid tumor	I	NCT00761696
	Metastatic pancreatic cancer	I/II: saridegib and gemcitabine	NCT01130142
	Advanced chondrosarcoma	II: saridegib or placebo	NCT01310816
Sonidegib	Advanced or metastatic hepatocellular carcinoma	I	NCT02151864
	Basal cell carcinoma	II: neoadjuvant sonidegib followed by surgery or imiquimod	NCT03534947
	Extensive stage small cell lung cancer	I: sonidegib, etoposide and cisplatin	NCT01579929
	Resectable pancreatic adenocarcinoma	I/II: sonidegib, gemcitabine, nab-paclitaxel in neoadjuvant setting	NCT01431794
	Localized prostate cancer	I	NCT02111187
	Multiple myeloma	II: sonidegib and lenalidomide	NCT02086552
	Esophageal cancer	I: sonidegib and everolimus	NCT02138929
	Advanced pancreatic cancer	I: sonidegib, fluorouracil, leucovorin, oxaliplatin, irinotecan	NCT01485744
	Pancreatic cancer	I/II: sonidegib, gemcitabine, and nab-paclitaxel	NCT02358161
	Advanced solid tumor	I	NCT01208831
	Advanced solid tumor	I	NCT00880308
	Solid tumors	I: sonidegib and paclitaxel	NCT01954355
	Advanced solid tumors	I: sonidegib and BKM120	NCT01576666
	Myeloid malignancies	I: sonidegib with azacytidine or decitabine	NCT02129101
	Advanced or metastatic basal cell carcinoma	II: sonidegib and buparlisib	NCT02303041
LEQ-506	Advanced solid tumors	I	NCT01106508
Taladegib	Advanced cancers	I	NCT01919398
	Esophageal cancer	I/II: Taladegib, paclitaxel, carboplatin, and radiation	NCT02530437
	Advanced solid tumors	I	NCT02784795
Glasdegib	Hematologic malignancies	I	NCT00953758
	Solid tumors	I	NCT01286467
	Acute myeloid leukemia	II	NCT01841333
	Hematologic malignancies	I: with standard chemotherapy agents	NCT02038777
	Acute myeloid leukemia	III: chemotherapy or azacytidine with or without glasdegib	NCT03416179
	Glioblastoma	I/II: glasdegib and temozolomide	NCT03466450
TAK-441	Advanced nonhematologic malignancies	I	NCT01204073
Vismodegib	Metastatic pancreatic adenocarcinoma	II: vismodegib, gemcitabine and nab-paclitaxel	NCT01088815
	Solid and hematologic malignancies	II: Canadian profiling and targeted agent utilization trial	NCT03297606
	Keratocystic odontogenic tumors	II	NCT02366312
	Acute myeloid leukemia	II: ribavirin, vismodegib with or without decitabine	NCT02073838
	Pancreatic adenocarcinoma	I: vismodegib and gemcitabine in neoadjuvant setting	NCT01713218
	Basal cell nevus syndrome, Gorlin syndrome	II	NCT00957229
	Breast cancer	II: neoadjuvant paclitaxel, epirubicin, cyclophosphamide with or without vismodegib	NCT02694224
	Glioblastoma	I/II: Neuro Master Match	NCT03158389
	Recurrent medulloblastoma	I	NCT00822458
	Metastatic pancreatic cancer or solid tumors	I: vismodegib, erlotinib, and gemcitabine	NCT00878163
	Advanced chondrosarcoma	Phase 1	NCT01267955
	Advanced basal cell skin cancer	I/II: pembrolizumab with or without vismodegib	NCT02690948
	Advanced solid tumors	II: My Pathway	NCT02091141
	Advanced head/neck basal cell carcinoma	II: vismodegib and radiation	NCT01835626
	Advanced gastric cancer	II	NCT03052478
	Multiple myeloma	I	NCT01330173
	Solid tumors, lymphomas or multiple myeloma	II: MATCH	NCT02465060
	Orbital and periocular basal cell carcinoma	IV	NCT02436408
	Medulloblastoma	II: with radiation and chemotherapy	NCT01878617
Vitamin D3	Basal cell carcinoma	I: with photodynamic therapy	NCT03483441
	Pancreatic cancer	III: high dose versus standard dose	NCT03472833
	Acute myeloid leukemia	II: deferasirox, vitamin D3, and azacitidine	NCT02341495
	Chronic lymphocytic leukemia, non-Hodgkin lymphoma	I	NCT02553447
	Indolent lymphoma	III: rituximab with or without vitamin D3	NCT03078855
Arsenic trioxide	High-grade glioma	I: with temozolomide and radiation therapy	NCT00720564
	Glioma	I: with radiation therapy	NCT00095771
	Glioma	I: stereotactic radiotherapy	NCT00185861
	Neuroblastoma and other childhood solid tumors	II	NCT00024258
	Basal cell carcinoma	I/II	NCT01791894

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
