# Peer review of "Recent Advances in the Clinical Targeting of Hedgehog/GLI Signaling in Cancer"

_cells, 2019, doi:10.3390/cells8050394_

Round 1

Reviewer 1 Report

In the manuscript “Recent Advances in the Targeting of Hedgehog Signaling in Cancer”, Xie et al. review contemporary hedgehog inhibitors in pre-clinical and clinical applications. In particular, the authors give a detailed overview about state-of-the-art hedgehog inhibitors in various cancers with a particular aim on different target sides. This is a well-structured paper with a high educational value. However, there are an abundance of writing issues that made portions of this manuscript difficult to read; the manuscript would be greatly aided by a more thorough editing process.

General comments:

In the abstract you mention explicit basal cell carcinoma. I would suggest keeping it more general and mention solid tumors to address a broader spectrum of readers.

The authors often refer to Hh ligands but there is no mention about the what these ligands are.

Figure 1 shows all three GLI family members, but there is no mention of these in the text.

Likewise, figure 1 shows GLIi and GLIa, neither of which are discussed in the manuscript.

I would suggest to only use abbreviations in the figure legend (Figure 1) since they have been introduced earlier

On line 25 the authors state there are 4 main components of the Hh signaling pathway, but only list 3 (PTCH1, SMO, and GLI).  Presumably the authors are referring to Hh as the 4th component, but this is not clearly articulated.

Would be helpful to have a statement about downstream target genes of Hh signaling as the authors refer to changes in GLI expression as evidence of Hh pathway inhibition throughout the review.

In the paragraph encompassing lines 67-75, the authors are either very vague in their description “ Hh pathway inhibition” or very specific (saridegib or vismodegib). It would be easier to read if the authors used the same level of depth for each of their examples.

The sentence “Pancreatic cancer stem cells in addition to desmoplasia were also suggested to contribute to chemotherapy resistance.” on lines 90-91 seems out of place.

Specific editing comments:

Lines 57-62 on page 2 should be rewritten to make the sentences less cumbersome to read.

The sentence on page 4, line 138 “In a mouse model of medulloblastoma….”  Should include the actual genetic alteration driving this mouse model.

On page 4, line 167 the authors make note of a study performed in Asian patients, however there is no mention of the country of origin of the “first-in-human trial” in the previous sentence.

There is no description of the clinical response of sonidegib mentioned on line 173.

Line 71 – Vismodegib has not yet been introduced

Line 159 – SCC has not been defined

A reference is needed on page 1, line 28.

A reference is needed on page 2, lines 68-69.

In the Abstract, line 11, should state “…and has been implicated…”

On page 1, lines 31-33 the authors write “Active SMO promotes the translocation of GLI family transcription factors into the nucleus and induces expression of target genes including cyclin-dependent kinases, and growth factors, ultimately promoting cell growth, survival and differentiation”. The comma between kinases and growth factors is not necessary.

Lines 67-68 should be changed to “…..decrease tumor growth in different….”

Line 73 on page 3 should have a comma after chondrosarcoma, not a period.

On page 3, line 88, the comma before “…increase gemcitabine delivery” should be replaced with an “and”.

Line 97 on page 3 should be “…in a mouse xenograft” or “…in mouse xenografts”

Line 116 on page 3 should be changed to ”…repurposing this anti-fungal as an anti-cancer agent”

On page 4, line 126 delete the phrase “It is known that in medulloblastoma”

On page 4, line 136 start a new paragraph before “In contrast to SMO inhibitors….”

Lines 144-145 on page 4 should read “….was able to effectively inhibit tumor growth in vivo.”

Please remove ‘as a result’ on page 5 line 225

Page 6, line 248 should read “…benefit in addition to standard chemotherapy…”

Reconsider the use of “hopes’” on page 11, line 320

Reword the sentence on line 338, page 11 to avoid the using “of not”.

Author Response

General comments:

In the abstract you mention explicit basal cell carcinoma. I would suggest keeping it more general and mention solid tumors to address a broader spectrum of readers.

Changes were made as suggested.

The authors often refer to Hh ligands but there is no mention about the what these ligands are.

Added as suggested.

Figure 1 shows all three GLI family members, but there is no mention of these in the text.

Added as suggested.

Likewise, figure 1 shows GLIi and GLIa, neither of which are discussed in the manuscript.

Added as suggested.

I would suggest to only use abbreviations in the figure legend (Figure 1) since they have been introduced earlier

Changes were made as suggested.

On line 25 the authors state there are 4 main components of the Hh signaling pathway, but only list 3 (PTCH1, SMO, and GLI).  Presumably the authors are referring to Hh as the 4th component, but this is not clearly articulated.

Changes were made as suggested.

Would be helpful to have a statement about downstream target genes of Hh signaling as the authors refer to changes in GLI expression as evidence of Hh pathway inhibition throughout the review.

They were mentioned in lines 33-34.

In the paragraph encompassing lines 67-75, the authors are either very vague in their description “ Hh pathway inhibition” or very specific (saridegib or vismodegib). It would be easier to read if the authors used the same level of depth for each of their examples.

Changes were made as suggested.

The sentence “Pancreatic cancer stem cells in addition to desmoplasia were also suggested to contribute to chemotherapy resistance.” on lines 90-91 seems out of place.

This is deleted as suggested.

Specific editing comments:

Lines 57-62 on page 2 should be rewritten to make the sentences less cumbersome to read.

Changes were made as suggested.

The sentence on page 4, line 138 “In a mouse model of medulloblastoma….”  Should include the actual genetic alteration driving this mouse model.

Added as suggested.

On page 4, line 167 the authors make note of a study performed in Asian patients, however there is no mention of the country of origin of the “first-in-human trial” in the previous sentence.

Added as suggested.

There is no description of the clinical response of sonidegib mentioned on line 173.

The trials discussed were phase I trials focusing on evaluating toxicity and tolerability.

Line 71 – Vismodegib has not yet been introduced

Changes were made as suggested.

Line 159 – SCC has not been defined

It should be BCC.

A reference is needed on page 1, line 28.

Added as suggested.

A reference is needed on page 2, lines 68-69.

This sentence was deleted.

In the Abstract, line 11, should state “…and has been implicated…”

Added as suggested.

On page 1, lines 31-33 the authors write “Active SMO promotes the translocation of GLI family transcription factors into the nucleus and induces expression of target genes including cyclin-dependent kinases, and growth factors, ultimately promoting cell growth, survival and differentiation”. The comma between kinases and growth factors is not necessary.

Deleted as suggested.

Lines 67-68 should be changed to “…..decrease tumor growth in different….”

Added as suggested.

Line 73 on page 3 should have a comma after chondrosarcoma, not a period.

Changes were made as suggested.

On page 3, line 88, the comma before “…increase gemcitabine delivery” should be replaced with an “and”.

Changes were made as suggested.

Line 97 on page 3 should be “…in a mouse xenograft” or “…in mouse xenografts”

Changes were made as suggested.

Line 116 on page 3 should be changed to ”…repurposing this anti-fungal as an anti-cancer agent”

Changes were made as suggested.

On page 4, line 126 delete the phrase “It is known that in medulloblastoma”

Changes were made as suggested.

On page 4, line 136 start a new paragraph before “In contrast to SMO inhibitors….”

Changes were made as suggested.

Lines 144-145 on page 4 should read “….was able to effectively inhibit tumor growth in vivo.”

Changes were made as suggested.

Please remove ‘as a result’ on page 5 line 225

Changes were made as suggested.

Page 6, line 248 should read “…benefit in addition to standard chemotherapy…”

Changes were made as suggested.

Reconsider the use of “hopes’” on page 11, line 320

Changes were made as suggested.

Reword the sentence on line 338, page 11 to avoid the using “of not”.

Changes were made as suggested.

Reviewer 2 Report

The following review delineates the current advancements in therapeutic interventions to target the Sonic Hedgehog pathway. Authors give a brief overview of Sonic hedgehog pathway and entail the underlying mutation in the pathway which leads to cancer. The review comprehensively covers the current landscape of drugs which are currently under pre-clinical and clinical trials for different types of cancer. Authors extensively cover current small molecules at various phases of clinical trials both alone and in combination. Only with minor grammatical corrections, the current review can be accepted in the present form.

Minor corrections:

Line 68 and 73 needs to be rephrased.

Author Response

Minor corrections:

Line 68 and 73 needs to be rephrased.

Changes were made as suggested.

Reviewer 3 Report

In this review, Xie et colleagues briefly describe the recent preclinical advances in targeting Hedgehog signaling in various cancers, summarizing some of the main results obtained in pre-clinical studies with inhibitors of the Hh pathway, and also focusing on both FDA-approved and experimental drugs. The manuscript presents useful tables that help the reader navigate data regarding clinical trials.

Despite the fact that this work does not describe all the recent advances and compounds developed to target Hh pathway (for example does not mention acylguanidine derivatives), this review is suitable for publication. There are, however, some minor concerns:

1)    Given the intense use of acronyms, a table listing all the abbreviations used in the manuscript could be beneficial to the reader.

2)    The paragraph “Ligand-dependent Hh signaling inhibition” would benefit from its division in sub-paragraphs, one for each compound that is presented.

3)    The citation [3] on line 29 does not seem the optimal reference for the statement. See

Alcedo, J., Ayzenzon, M., Von Ohlen, T., Noll, M. and Hooper, J. E. (1996). The Drosophila smoothened gene encodes a seven-pass membrane protein, a putative receptor for the Hedgehog signal. Cell 86, 221-232. 

Alcedo, J. and Noll, M. (1997). Hedgehog and its Patched-Smoothened receptor complex: a novel signalling mechanism at the cell surface. Biol. Chem. 378, 583-590. 

Chen, Y. and Struhl, G. (1996). Dual roles for patched in sequestering and transducing Hedgehog. Cell 87, 553-563. 

Chen, Y. and Struhl, G. (1998).In vivo evidence that Patched and Smoothened constitute distinct binding and transducing components of a Hedgehog receptor complex. Development 125, 4943-4948 (1998)

4)   Line 69 lacks a reference for Hh inhibition in SCLC.

5)   Line 71 please mention what Vismodegib is, since it is the first time that Vismodegib appears in the text.

6)   Lines 89-94 the sentences don’t flow organically: do the authors imply that vitamin D3 is related to prancreatic stem cells? 

7)   Line 101 what do the authors mean for “chemically different”? How does Saridegib differ from the other drugs? 

8)   Please provide all the IC50s of the drugs mentioned in paragraph 2 “ligand-dependent Hh signaling inhibition”, since it has been done for half of the compunds described.

9)   A brief introduction of what “ligand-independent Hh signaling inhibition” entails without necessarily being connected to medulloblastoma (since mutations of Hh pathway genes are found in a variety of tumors) would be a better start of the paragraph. 

10)  As for the previous paragraph, the paragraph “Ligand-independent Hh signaling inhibition” would benefit as well from a re-organization in sub-paragraphs, one for each compound that is presented. 

11)  Line 152 “both mutations” might sound better than only “both”.

12)  The sentences in lines 168-172 and 175-182 are not very clear and would benefit from being rewritten.

13)  Line 191 since it is a different trial, starting a new line for the Vismogenib trial would be helpful.

14)  Line 203 perhaps the authors meant “studies” instead of “study”

15)  Line 211 “the findings…were confirmed” (not “was”)

16)  Line 331 please place the references next to each statement they are related to, and not at the end of the five points.

Author Response

1)    Given the intense use of acronyms, a table listing all the abbreviations used in the manuscript could be beneficial to the reader.

We would be happy to provide a list of all abbreviations if the journal format allows.

2)    The paragraph “Ligand-dependent Hh signaling inhibition” would benefit from its division in sub-paragraphs, one for each compound that is presented.

Changes were made as suggested.

3)    The citation [3] on line 29 does not seem the optimal reference for the statement. See

Alcedo, J., Ayzenzon, M., Von Ohlen, T., Noll, M. and Hooper, J. E. (1996). The Drosophila smoothened gene encodes a seven-pass membrane protein, a putative receptor for the Hedgehog signal. Cell 86, 221-232.

Alcedo, J. and Noll, M. (1997). Hedgehog and its Patched-Smoothened receptor complex: a novel signalling mechanism at the cell surface. Biol. Chem. 378, 583-590. 

Chen, Y. and Struhl, G. (1996). Dual roles for patched in sequestering and transducing Hedgehog. Cell 87, 553-563. 

Chen, Y. and Struhl, G. (1998).In vivo evidence that Patched and Smoothened constitute distinct binding and transducing components of a Hedgehog receptor complex. Development 125, 4943-4948 (1998)

 Changes were made as suggested.

4)   Line 69 lacks a reference for Hh inhibition in SCLC.

The sentence was deleted.

5)   Line 71 please mention what Vismodegib is, since it is the first time that Vismodegib appears in the text.

Changes were made as suggested.

6)   Lines 89-94 the sentences don’t flow organically: do the authors imply that vitamin D3 is related to prancreatic stem cells? 

No. Changes were made as suggested.

7)   Line 101 what do the authors mean for “chemically different”? How does Saridegib differ from the other drugs? 

Changes were made as suggested. It meant structurally different.

8)   Please provide all the IC50s of the drugs mentioned in paragraph 2 “ligand-dependent Hh signaling inhibition”, since it has been done for half of the compunds described.

Experiments in the reference were done mainly in xenografts. No IC50 value was provided.

9)   A brief introduction of what “ligand-independent Hh signaling inhibition” entails without necessarily being connected to medulloblastoma (since mutations of Hh pathway genes are found in a variety of tumors) would be a better start of the paragraph. 

Changes were made as suggested.

10)  As for the previous paragraph, the paragraph “Ligand-independent Hh signaling inhibition” would benefit as well from a re-organization in sub-paragraphs, one for each compound that is presented. 

Changes were made as suggested.